# From Assistant to Independent Developer — Are GPTs Ready for Software Development?

**Dezhi Ran**[1,2]**, Yuan Cao**[1,2]**, Mengzhou Wu**[1,2]**, Simin Chen**[3]**, Yuzhe Guo**[1,2]**,**
**Jun Ren**[4]**, Zihe Song**[5]**, Hao Yu**[6]**, Jialei Wei**[2,4]**, Linyi Li**[7]**, Wei Yang**[2,4]**,**
**Baishakhi Ray**[3]**, Tao Xie**[1,2,4]
Correspondence to: taoxie@pku.edu.cn

[1]Peking University    [2]Beijing Tongming Lake Center    [3]Columbia University
[4]Fudan University    [5]University of Texas at Dallas
[6]Hong Kong University of Science and Technology    [7]Simon Fraser University

## Abstract

Large language models (LLMs) have demonstrated remarkable capability in function-level code generation tasks. Unlike isolated functions, a real-world application demands reasoning over the entire software system: developers must orchestrate how different components interact, maintain consistency across states over time, and ensure that the application behaves correctly within the lifecycle and framework constraints. Yet, no existing benchmark adequately evaluates whether LLMs can bridge this gap and construct entire software systems from scratch.

To address this gap, we propose APPFORGE, a benchmark consisting of 101 software development problems drawn from real-world Android apps. Given a natural language specification detailing the app functionality, an LLM is tasked with **implementing the functionality into an Android app from scratch**. Developing an Android app from scratch requires understanding and coordinating app states, lifecycle management, and asynchronous operations, calling for LLMs to generate context-aware, robust, and maintainable code. To construct APPFORGE, we design a multi-agent system to automatically summarize the main functionalities from app documents and navigate the app to synthesize test cases validating the functional correctness of app implementation. Following rigorous manual verification by Android development experts, APPFORGE incorporates the test cases within an automated evaluation framework that enables reproducible assessment without human intervention, making it easily adoptable for future research. Our evaluation on 12 flagship LLMs show that all the evaluated models achieve low effectiveness, with the best-performing model (GPT-5) developing only 18.8% functionally correct applications, highlighting fundamental limitations in current models' ability to handle complex, multi-component software engineering challenges. Our benchmark, including all data and code, is publicly available at https://github.com/TongmingLAIC/AppForge.

## 1 Introduction

Large language models (LLMs) are reshaping the horizon of software engineering. Frontier code LLMs (OpenAI, 2023) are deeply integrated into developer's toolchains such as GitHub Copilot (GitHub, 2025), Amazon CodeWhisperer (Amazon Web Services, 2025), and Claude Code (Anthropic, 2025). They are advancing from coding assistants to fully autonomous software developers (Yang et al., 2024), which hold significant potential to shape the next generation of software engineering.

Although existing benchmarks have advanced the evaluation of code LLMs, they primarily focus on generating isolated snippets or functions (Chen et al., 2021); such generation differs fundamentally from the system-level reasoning and integration required to build a complete application. As a result, they cannot determine whether current models are capable of end-to-end software development in

real-world scenarios. For instance, HumanEval focuses on self-contained, toy-level, function-level code generation, while SWE-Bench (Jimenez et al., 2024) targets program repair tasks within an existing codebase, requiring only minor modifications to a few lines of code in the target repository. None of the existing benchmarks effectively assess the end-to-end software development capabilities of LLMs in the role of an independent software developer (Liu et al., 2023; Jain et al., 2024; White et al., 2025; Zhu et al., 2024; Rajore et al., 2024). To address the limitations of current benchmarks and evaluate whether LLMs can truly function as software engineers in real-world development scenarios, we argue for the creation of a new benchmark that goes beyond narrow tasks and instead captures the full spectrum of software development.

Building such a benchmark is necessary because (1) it provides a comprehensive and realistic evaluation of LLMs' ability to perform software development tasks end-to-end, (2) it bridges the gap between isolated code generation and real-world engineering, and (3) it provides insight how to leverage LLMs for the next generation software engineering. However, there are three main challenges in building such a benchmark. ❶ *Reflecting the Real-World Software Development Process*. The benchmark should be realistic and faithfully represent the complexities and workflows of actual software development. ❷ *Ensuring Sufficient Challenge and Diversity*. The benchmark should be sufficiently challenging to differentiate model capabilities, covering diverse tasks such as design, implementation, debugging, and maintenance. ❸ *Measuring End-to-End Development Performance*. The benchmark should capture not only code correctness but also factors such as code quality, maintainability, and integration within larger systems.

To address these challenges, we propose Android application (app) development as our benchmark domain, motivated by three key factors. First, Android represents one of the most significant software ecosystems globally, with over 2.6 million apps available (Technource, 2022), making it highly representative of real-world software development. Android development naturally involves creating complete projects with specific functional requirements, effectively capturing authentic development workflows. Second, developing Android apps from scratch provides inherent complexity through backend logic implementation, state management, UI design, and external API integration, ensuring sufficient difficulty and diversity for comprehensive evaluation. Third, the mature ecosystem of Android development tools, including static analyzers, testing frameworks, and emulation environments, enables rigorous automated assessment of various development aspects (Developers, 2025a;b;c).

Building on this intuition, we propose APPFORGE, the first benchmark for evaluating code LLMs specifically in Android app development. As illustrated in Figure 2, LLMs are tasked with generating complete Android apps from scratch based on natural language specifications. Once the code files are generated, APPFORGE automatically handles compilation into APK files, deployment on Android emulators, and comprehensive functionality validation against automated test case execution and systematic fuzzing. To ease the use of APPFORGE, the evaluation of APPFORGE is fully automated and encapsulated with a standalone docker for out-of-the-box usage.

To construct APPFORGE with scalability and rigor, we first collect real-world Android apps from F- Droid (Project, 2025c), a well-curated repository of open-source Android apps for providing real-world and actively maintained projects. Next, we leverage LLMs to automatically extract and summarize functionality specifications from each app's documentation and source code. Subsequently, we leverage a GUI agent (Ran et al., 2024) to interact with the app, capturing its runtime behavior to validate and enrich the specification description to avoid task ambiguity. Finally, we engage Android development experts to verify the correctness of both specifications and synthesized test cases. This combination of automated processing and expert validation ensures both scalability and reliability in benchmark construction.

We evaluate 12 flagship LLMs (OpenAI, 2023; Guo et al., 2024; Di et al., 2024; Jiang et al., 2024; Li et al., 2022) including GPT-5 (OpenAI, 2025) and Claude-4-Opus (Anthropic, 2025) as well as popular coding agents including Claude Code (Anthropic, 2025) on APPFORGE, revealing three key findings. First, all models achieve remarkably low performance with less than 20% of apps being functionally correct, and among these correct apps, half still encounter at least one crash (detailed in Table 1). This result contrasts sharply with existing saturated benchmarks (Chen et al., 2021; Jain et al., 2024; White et al., 2025), indicating a significant gap between current LLM capabilities and real-world development tasks, and that APPFORGE represents the next frontier of software engineering challenges. Second, we uncover that some LLMs evade app development tasks by sacrificing functionality integrity for compilation success. When given opportunities to improve their

previous generations with compilation errors, GPT-4.1 (OpenAI, 2025) and Kimi-K2 (Kimi Team, 2025) delete the implementation of error-inducing functions instead of fixing them as illustrated in Figure 8, indicating an avoidance strategy that sidesteps error handling instead of demonstrating true debugging competence. Specifically, GPT-4.1 evades development in 91.09% of tasks, while Kimi K2 does so in 65.36% of tasks. Finally, for simple tasks such as calculator implementation (Figure 6), LLMs demonstrate promising performance, producing robust apps that surpass typical human-written code quality as illustrated in Figure 7, suggesting significant potential when complexity is appropriately managed for future software development.

We summarize our main contribution as follows:

- *New real-world problem*. We introduce end-to-end Android app development from scratch as a comprehensive evaluation task for LLMs' software engineering capabilities.

- *New Benchmark*. We construct APPFORGE, a benchmark with 101 diverse Android development tasks and fully automated evaluation suites.

- *Evaluation Results*. We evaluate 12 flagship LLMs and analyze their limited performance and failure patterns in real-world software engineering scenarios.

## 2 BACKGROUND & RELATED WORK

Code LLMs have been advancing rapidly, where frontier code LLMs such as GPT5 (OpenAI, 2025), Claude-Opus (Anthropic, 2025), Gemini-Pro (Google, 2025), and Qwen3-Coder (Yang et al., 2025) have reshaped the paradigm of software development. As software ecosystems such as Cursor (Cursor, 2025) and GitHub Copilot (GitHub, 2025) continue to mature, the application scenarios of code LLMs expand beyond code generation and completion to encompass debugging, test generation, and even autonomous software development.

In contrast to the wide application scenarios of code LLMs, benchmarks that evaluate code LLMs still largely focus on (1) function-level code generation and completion, such as HumanEval (Chen et al., 2021), MBPP (Austin et al., 2021), and BigCodeBench (Zhuo et al., 2024); and (2) patch generation and feature implementation with repository-level context, such as SWE-Bench (Jimenez et al., 2024), Web-bench (Xu et al., 2025) and Lo-CoBench (Qiu et al., 2025). Some efforts transform static benchmarks into dynamic ones to combat data contamination, such as SWE-Bench-Live (Zhang et al., 2025) and LiveCodeBench (Jain et al., 2024). As shown in Figure 1, APPFORGE goes beyond function-level code generation and patch generation. Compared to existing benchmarks, APPFORGE evaluates code LLM's capability to perform **automated software development from scratch** at the

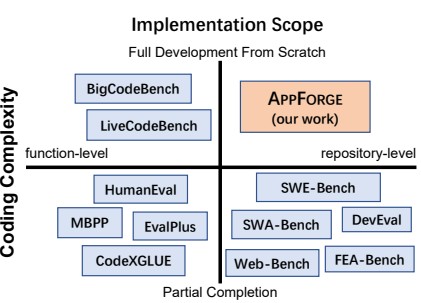

Figure 1: Our Work Compared with Existing Code Generation Benchmarks.

repository level. It incorporates rigorous evaluation empowered by automated test cases and systematic fuzzing.

APPFORGE is the first benchmark for assessing LLM capabilities in Android development to our knowledge. Android apps are typically built with Java or Kotlin following Material Design and Android architecture guidelines, and comprise multiple interconnected components (2025a). Android apps represent one of the most significant software ecosystems globally with over 2.6 million apps available (Technource, 2022), so we believe that Android development is an ideal code LLM evaluation scenario that largely reflects the real-world software development process. F-Droid (2025c) is the leading open-source Android app repository, serving millions of users worldwide. F-Droid apps span diverse categories and their code undergoes rigorous review process. APPFORGE is constructed from a diverse set of high-quality F-Droid apps, and can be dynamically expanded using latest projects from F-Droid. We defer a more in-depth discussion of related work in Appendix A.

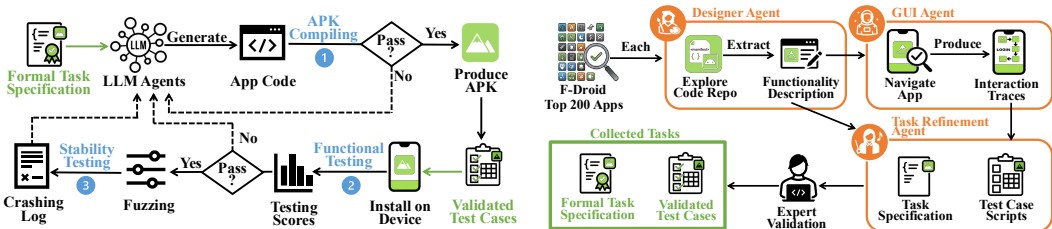

Figure 2: Workflow of APPFORGE.          Figure 3: Construction of APPFORGE.

# 3  APPFORGE

APPFORGE is a benchmark designed to evaluate LLMs' capabilities across the full software development lifecycle for Android apps, using real-world apps such as Amaze File Manager (F-Droid, 2025a), Arcticons (F-Droid, 2025b), and Vanilla Music (F-Droid, 2025c). Given the natural language description of an Android app, the task is to generate the corresponding code implementation that not only faithfully realizes the described functionality and passes the associated tests, but also executes securely within the Android operating system.

## 3.1  ANDROID APP DEVELOPMENT TASK FORMULATION

Each task in APPFORGE includes three main fields: *model input*, *model output*, and the *evaluation suite*. An example of task instance is provided in Appendix B.1.

**Model Input**: The model input is a natural language description that consists of three components: (1) a high-level overview of the app's functionality along with detailed descriptions of the features corresponding to each functionality, (2) natural language test cases that specify how these functionalities should be implemented and validated, and (3) implementation constraints such as API version requirements and expected output format specifications. Within the detailed feature descriptions, we also provide the specific resource IDs required for implementation. This design streamlines the overall evaluation process.

**Expected Model Output**: When prompting the LLM for app generation, the model is required to produce output in the JSON format, where each key represents a filename and each value contains the corresponding code. This design enables automated project assembly and evaluation.

**Evaluation Suite**: APPFORGE includes an automated evaluation pipeline consisting of three components: (1) an automatic compiler suite that parses the generated outputs, assembles them into an Android project, and compiles the project into an Android Package (APK); (2) a testing module that installs the APK onto an Android emulator and executes predefined test cases to validate functional correctness; and (3) a lightweight fuzzer that evaluates the robustness and exception-handling capabilities of the app under various edge cases and unexpected inputs. The evaluation reports four metrics: (1) compilation success rate, (2) test pass rate, (3) crash rate, and (4) an overall performance score (detailed in Appendix B.3) representing the model's effectiveness on the given Android development benchmark (More implementation details could be found in Appendix B.4).

## 3.2  CONSTRUCTION OF APPFORGE

We construct our benchmark from real-world Android apps collected from F-Droid. Although each app on F-Droid comes with detailed documentation and README files, these resources are too large and unstructured to be directly used as prompts for benchmarking LLMs. To address this limitation, we need to regenerate the task. Specifically, we follow the pipeline below: (1) Seed app selection: we choose apps based on diversity, complexity, and popularity. (2) UI navigation and trace recording: we use a UI navigator tool to explore the selected apps and record the navigation traces along with each UI element's ID. This step provides detailed interaction data, enabling automatic evaluation. Since our UI navigation is a dynamic process, even the same app can produce different traces. This dynamic mechanism allows us to generate diverse tasks from the same app, reducing the risk of data contamination. (3) Trace summarization: we combine the app documentation and navigation

traces, such as the element ID, and then use an LLM to summarize each trace into natural language descriptions. (4) Human validation: finally, we perform human validation to ensure that generated tasks are accurate and meaningful.

**App Selection and Scraping.** We begin by ranking apps based on a combination of popularity, complexity, diversity, and update frequency; detailed criteria are in Appendix B.5. From this ranking, we select the top 200 highest-scoring apps across different categories as seed apps for subsequent task creation, ensuring balanced coverage of Android development domains. For each selected app, we analyze its code repository to extract metadata, including descriptions from README files and release notes. We then summarize the app's core functionalities in natural language using the JSON format. These functionality descriptions are intentionally high-level and may be ambiguous.

**Automatic App Navigation.** We use an existing tool, UIAutomator (Android Developers, 2025), to install each seed app in an Android emulator and systematically record interaction traces. For every high-level functionality description, a UI navigator performs goal-based navigation (Ran et al., 2024) starting from the app's main screen. Guided by the functionality description, the agent identifies and interacts with relevant UI elements while maintaining a detailed log of the process. At each step, it captures the full UI tree using UIAutomator, including element properties such as text, resource-id, class, and bounds. The agent also documents the sequence of UI actions (e.g., clicks, text inputs, swipes), the target elements, and the reasoning behind each action, along with the resulting screen transitions and state changes. Once the target functionality is accomplished (e.g., logging in or sending a message), the agent records the complete interaction trace. This goal-directed approach produces precise traces that capture the most natural paths for implementing each functionality.

**Task Generation For Trace History.** We then utilize an LLM to synthesize precise task descriptions and test suites based on the captured interaction traces. First, the task refinement agent transforms each interaction trace into a test case. Each test case consists of a sequence of UI actions and an oracle specifying the expected outcome of executing the action sequence. Each UI action is associated with a UI element containing clear text or resource-id labels. For UI elements in seed apps that lack meaningful labels, the refinement agent generates context-appropriate resource-ids to avoid ambiguity. Each oracle is an assertion determining whether a UI element exists or does not exist. The test case is implemented as a Python script using the UIAutomator framework, enabling automated evaluation. Based on the synthesized test suites, the task refinement agent generates a task description detailing the core functionalities and their implementation. For each test script, the agent produces natural language descriptions that specify the sequence of UI interactions (e.g., "click the button with login resource-id", "enter text in the username field") and the expected app states after these operations. This approach eliminates ambiguity by providing precise, actionable specifications that ensure any LLM or human developer interpreting the task description will implement functionally equivalent apps that satisfy the same behavioral requirements.

**Android Developer Validation.** To ensure quality control, five expert Android developers with a combined 30 years of experience review all tasks for technical accuracy, feasibility, and alignment with real-world practices. The validation process includes checking task clarity and completeness, verifying non-trivial and unambiguous requirements, ensuring coverage of essential concepts across difficulty levels, and confirming the soundness of examples and constraints. Experts also validate test cases by examining expected outputs and the accuracy of automated testing. Each task undergoes multiple review rounds until consensus is reached, yielding high-quality benchmarks that reflect authentic Android development challenges.

## 3.3 BENCHMARK SUITE AND DATA STATISTICS

We collect 101 high-quality Android development tasks, each representing the development of a complete Android app. The task distribution reflects real-world Android development patterns and emphasizes comprehensive app diversity: UI/Layout focused apps comprise 40%, covering complex view hierarchies, custom components, and responsive design; Backend Integration apps account for 32%, including API consumption, data persistence, and background services; User Interaction apps represent 94%, focusing on gesture handling, input validation, and navigation flows; and System Integration apps make up 63%, encompassing permissions, hardware access, and inter-app communication. Task complexity spans three difficulty levels based on implementation requirements: Beginner (37%, focusing on single-activity apps with basic Android concepts), Intermediate (48%, requiring

multi-component integration and moderate architectural complexity), and Advanced (15%, involving sophisticated architectural patterns, performance optimization, and complex system interactions). The diversity in app categories and complexity levels ensures that APPFORGE captures the full spectrum of Android development scenarios in real-world practice.

### 3.4 KEY FEATURES OF APPFORGE

Traditional code generation benchmarks often focus on toy function-level tasks or partial repository generation, where much of the context is pre-defined and evaluation is limited to functionality—an approach shown to be insufficiently rigorous in prior work. In contrast, APPFORGE draws on real-world Android apps from F-Droid, offering authentic, end-to-end development tasks that more faithfully capture practical software engineering challenges. Here, we describe some key features:

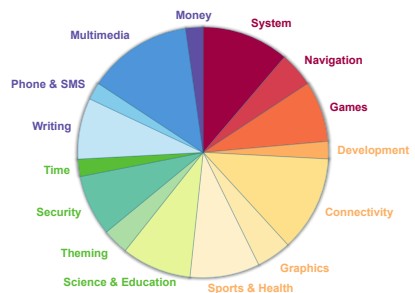

Figure 4: Distribution of Category.

**Real-world Software Development Tasks.** Since each task in APPFORGE is sourced from F-Droid and represents a real-world Android app that may have been installed on millions of devices worldwide, solving APPFORGE requires LLMs to demonstrate sophisticated skills and knowledge in full-stack Android development, including UI design, API integration, state management, and security considerations—capabilities rarely evaluated in traditional code generation benchmarks.

**Diverse Task Categories.** As shown in Figure 4, APPFORGE includes a diverse range of apps. Each instance of APPFORGE belongs to a unique category, making it significantly more diverse than existing benchmarks (e.g., SWE-Bench includes only 12 different repositories from Python (Jimenez et al., 2024) and concurrent work LoCoBench covers only 3 mobile app categories (Qiu et al., 2025)).

**Software-level Code Generation.** This task challenges LLMs to generate coherent, end-to-end Android app code while understanding the semantics of APIs across different versions of the Android framework and third-party libraries. Unlike function-level tasks, software-level generation requires reasoning about how components interact, handling version-specific behavior, and integrating multiple modules correctly. By requiring models to adapt to evolving APIs and manage compatibility, this task evaluates a deeper level of software engineering capability, beyond simple functionality, ensuring that generated apps are both correct and maintainable.

**Rigorous Functionality & Reliability Evaluation.** Considering the fact that every software may contain some bug or defect, our benchmark includes both functionality and reliability evaluations. Our experiments demonstrate that incorporating reliability is essential, as it can uncover hidden crashes that would be missed by functionality testing alone.

**Wide Solution Space.** The task of full-app code generation in APPFORGE provides a level playing field for evaluating approaches ranging from standard models to autonomous agents capable of reasoning and acting across an entire Android project. APPFORGE also encourages creative solutions, allowing models to produce implementations that may diverge from reference apps while still meeting functional and security requirements.

## 4 EVALUATIONS

### 4.1 EVALUATION SETUP

We conduct comprehensive experiments on APPFORGE with 12 state-of-the-art LLMs, including 7 proprietary models (Claude-5-Opus, Claude-4-Sonnet(Anthropic, 2025), Gemini-2.5-Pro (Google, 2025), GPT-4.1 (OpenAI, 2025), GPT-5-Low, GPT-5-Medium, and GPT-5-High (OpenAI, 2025)) and 5 open-source models (DeepSeek-R1, DeepSeek-V3 (Guo et al., 2024), GLM-4.5 (Zhuo et al., 2024), Kimi K2 (Kimi Team, 2025), and Qwen3-Coder (Yang et al., 2025)), along with two coding agents (mini-SWE-agent (Yang et al., 2024) and Claude Code (Anthropic, 2025)) to evaluate the cutting-edge progress in fully automated software engineering. Details are provided in Appendix C.

Table 1: Performance of LLMs on APPFORGE.

| LLMs | Pass@1 | | | | | | with Compilation Error Feedback | | | | | |
|---|---|---|---|---|---|---|---|---|---|---|---|---|
| | #File | #LOC | Compile | Test Pass | Crash | Success | #File | #LOC | Compile | Test Pass | Crash | Success |
| *Proprietary Models* | | | | | | | | | | | | |
| Claude-4-Opus | 9.11 | 396.94 | **80.20%** | **28.52%** | 60.49% | 11.88% | 8.97 | 386.63 | **90.10%** | 34.22% | 60.44% | 14.85% |
| Claude-4-Sonnet | 9.61 | 432.17 | 40.59% | 10.35% | 58.54% | 0.99% | 9.78 | 437.69 | 77.23% | 18.36% | 26.92% | 3.96% |
| Gemini-2.5-Pro | 10.74 | 380.31 | 53.47% | 19.63% | 62.96% | 7.92% | 10.52 | 361.94 | 68.32% | 21.63% | 75.36% | 13.86% |
| GPT-5-High | 7.76 | 354.59 | 45.54% | 21.90% | 52.17% | **14.85%** | 7.36 | 340.77 | 82.18% | 29.07% | **31.33%** | **18.81%** |
| GPT-4.1 | 8.00 | 367.43 | 6.93% | 2.44% | **28.57%** | 0.99% | 2.68 | 58.41 | 74.26% | 1.85% | 94.67% | 0.99% |
| *Open-source Models* | | | | | | | | | | | | |
| DeepSeek-R1 | 7.00 | 214.33 | 14.85% | 1.90% | 73.33% | 0.00% | 7.33 | 233.78 | 44.55% | 12.29% | 62.22% | 4.95% |
| DeepSeek-V3 | 5.17 | 164.67 | 5.94% | 2.23% | 83.33% | 0.99% | 5.33 | 250.19 | 26.73% | 10.40% | 48.15% | 4.95% |
| GLM-4.5 | 7.64 | 256.16 | 24.75% | 8.74% | 72.00% | 4.95% | 8.51 | 278.91 | 44.55% | 10.14% | 75.56% | 4.95% |
| Kimi K2 | 6.82 | 239.82 | 16.83% | 4.95% | 76.47% | 1.98% | 5.10 | 168.60 | 41.58% | 7.76% | 69.05% | 1.98% |
| Qwen3-Coder | 5.29 | 209.00 | 27.72% | 4.42% | 75.00% | 1.98% | 6.20 | 241.21 | 85.15% | 21.45% | 29.07% | 8.91% |

## 4.2 MAIN RESULTS

**All models struggle on APPFORGE.** As shown in Table 1, all models achieve low performance on APPFORGE, with the best-performing flagship model GPT-5 with high reasoning mode achieving only 14.85% success rate (developing 14.85% of apps passing all test cases). When given chances to repair compilation errors in their previous development, the improvement is still marginal, with GPT-5 achieving only 18.81% success rate. Open-source models perform considerably worse, all achieving less than 10% functional success rate after repairing compilation errors. While the high compilation rates of flagship models demonstrate that existing models can generate syntactically correct programs, the consistently low test pass rates across all models reveal the fundamental challenge of generating functionally correct Android apps. In addition, over 50% of functionally correct apps crash during runtime, highlighting that even when LLMs successfully implement the required functionality, the generated code often lacks reliability necessary for real-world deployment.

**Iterative refinement with compilation feedback does not significantly improve functional correctness.** The compilation error feedback substantially improves compilation success across all models, with notable improvements for Claude-4-Sonnet (40.59% to 77.23%) and Qwen3-Coder (27.72% to 85.15%). However, this improvement does not translate proportionally to functional correctness, as test pass rates show modest gains. As illustrated in Figure 5, iterative refinement significantly improves compilation success for both Qwen3-Coder (33.7% to 98%) and DeepSeek-V3 (7.9% to 63.4%). However, the functional success rate, measured by passing test cases, saturates quickly after 2-3 iterations, peaking around 23% for Qwen3-Coder and 14% for DeepSeek-V3.

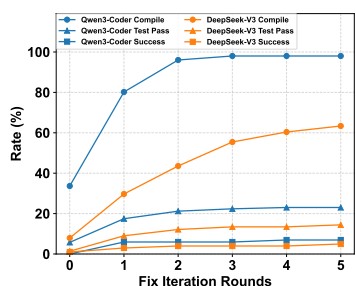

Figure 5: Performance evolution with compilation feedback.

**LLMs can develop robust, functionally correct apps on simple development tasks.** Despite overall low success rates, successful cases demonstrate that LLMs can generate surprisingly sophisticated Android apps. As visualized in Figure 6, there is a clear inverse relationship between app complexity and success rates for lower complexity tasks with enough sample sizes below 800 LOCs. Notably, successful cases often showcase proactive exception handling and defensive programming beyond basic functional requirements. Figure 7 illustrates an actual implementation by GPT-5 in the Autostarts app, where it gracefully manages potential exceptions and provides fallback solutions.

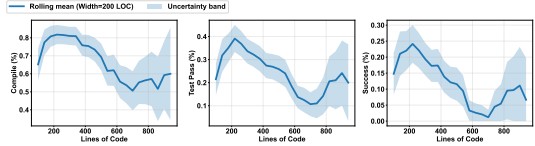

Figure 6: Correlation between Lines of Code (LOC) and evaluation metrics (Compile, Test Pass, and Success). Rolling means with uncertainty bands show performance variability across code complexity.

**Some LLMs evade development tasks rather than repairing their compilation errors.** Interestingly, GPT-4.1 and Kimi K2 evade development tasks during iterative refinement, where they delete

```
1  private void openAppInfo(String packageName) {
2    try {
3      Intent it = new Intent(Settings.ACTION_APPLICATION_DETAILS_SETTINGS);
4      it.setData(Uri.parse("package:" + packageName));
5      it.addFlags(Intent.FLAG_ACTIVITY_NEW_TASK);
6      startActivity(it);
7    } catch (ActivityNotFoundException e) {
8      try {
9        Intent fallback = new Intent(Settings.ACTION_APPLICATION_DETAILS_SETTINGS);
10       fallback.setData(Uri.parse("package:" + getPackageName()));
11       startActivity(fallback);
12     } catch (Exception ex) {
13       Toast.makeText(this, "Unable to open Application Info", Toast.LENGTH_SHORT).show();
14     ......
```

Figure 7: Proactive defensive programming implemented by GPT-5 on the Autostarts app.

```
1  // --- Original Geneartion (Compile Error: MainActivity.java:45: error: cannot find symbol
       Intent it = new Intent(MainActivity.this, NewTodoListActivity.class);) ---
2  findViewById(R.id.ac_add).setOnClickListener(v -> {
3    Intent it = new Intent(MainActivity.this, NewTodoListActivity.class);
4    startActivityForResult(it, 101);
5  });
6  // --- Refinement (passing compilation without implementation) ---
7  findViewById(R.id.ac_add).setOnClickListener(v -> {
8  });
```

Figure 8: GPT-4.1 evades development when fixing the compilation error on Todo List app.

faulty implementations instead of repairing them. GPT-4.1 shows a dramatic reduction in generated number of files (from 8.00 to 2.68) and LOCs (from 367.43 to 58.41) when provided with compilation feedback. As shown in Figure 8, GPT-4.1 replaces the buggy implementation of the function with an empty body. This strategy successfully achieves the highest compilation rate improvement (from 6.93% to 74.26%), but does no good for implementing the required app functionality. Similar patterns are observed in Kimi K2, indicating that some LLMs may strategically simplify their solutions when faced with compilation challenges rather than addressing the underlying issues.

**APPFORGE differentiates model capabilities better than existing code generation benchmarks.** While many models achieve high and similar performance on traditional code generation benchmarks such as HumanEval and SWE-bench, APPFORGE helps reveal performance gaps of LLMs for real-world software engineering tasks with success rates spanning from 0.99% to 14.85%. In addition, We visualize the model performance differentiation in Figure 9. the performance variance on APPFORGE is substantially larger than SWE-bench, providing more nuanced dif-

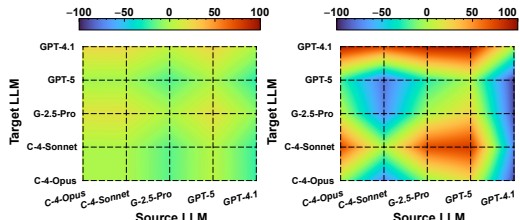

Figure 9: Pairwise relative performance differences between models on SWE-bench-verified (**Left**) and APP-FORGE (**Right**). Green and red cells represent relatively superior and inferior performance, respectively, with color intensity indicating the magnitude of differences.

ferentiation of model capabilities. This result suggests that APPFORGE captures the real-world software engineering challenges that are not adequately captured by previous code generation benchmarks.

## 4.3 PERFORMANCE OF ADVANCED CODING AGENTS AND REASONING MODELS

Table 2: Performance of coding agents on APPFORGE.

| Agent | LLM | #File | #LOC | Compile | Test Pass | Success |
|-------|-----|-------|------|---------|-----------|---------|
| SWE | Claude-4-Opus | 10.76 | 558.40 | 71.29% | 24.61% | 11.88% |
| | Qwen3-Coder | 8.42 | 430.94 | 88.12% | 22.21% | 6.93% |
| CC | Qwen3-Coder | 5.34 | 280.66 | 76.24% | 14.64% | 6.93% |

Table 3: Performance of GPT-5 with different reasoning levels on APPFORGE.

| Level | #File | #LOC | Compile | Test Pass | Success |
|-------|-------|------|---------|-----------|---------|
| Low | 5.91 | 280.91 | 22.77% | 8.41% | 2.97% |
| Medium | 7.61 | 321.96 | 27.72% | 11.11% | 3.96% |
| High | 7.76 | 354.59 | 45.54% | 21.90% | 14.85% |

**Coding agents provide marginal improvements at substantial computational cost.** As shown in Table 2, coding agents (mini-SWE-agent as SWE, Claude Code as CC) exhibit slight improvements over simpler baseline approaches, and yet their performance gains are modest. Specifically, the best-performing combination (mini-SWE-agent using Claude-4-Opus) achieves only an 11.88% functional success rate. Although these agents demonstrate potential for iterative refinement and error correction, their modest overall performance indicates that current agent-based frameworks still fall short of effectively overcoming critical challenges inherent to real-world software engineering tasks, such as multi-file integration and framework-specific complexity typical in Android app development.

**Enhancing reasoning capabilities remain insufficient for Android development.** Table 3 demonstrates that increasing the reasoning level of GPT-5 leads to improved performance across all metrics, with the highest reasoning setting achieving 14.85% functional success compared to 2.97% at the low level. However, even with maximum reasoning enhancement, the absolute performance remains far from satisfactory for practical Android development, highlighting that the fundamental challenges of multi-file coordination, framework-specific knowledge, and complex dependency management require more than enhanced reasoning alone.

## 4.4 ANALYSIS OF DEVELOPMENT CHALLENGES

**Compilation Error Analysis.** Figure 10 presents the distribution of compilation errors. The most prevalent error stems from "Android Resource Linking Failed", accounting for 39.7% of compilation errors. This compilation error is typically caused by missing or misreferenced resource files in the generated apps, highlighting current models' inadequate capability in comprehensive software engineering tasks that require systematic coordination across multiple project components. An interesting observation

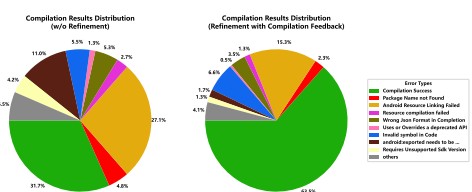

Figure 10: Distribution of compilation errors across generated Android apps.

is that GPT series models and Kimi-K2 encounter the issue that apps fail to compile due to missing android:exported declarations (an attribute requirement introduced in Android 12 (Android Developers, 2021)), highlighting a gap between LLM training data and current Android requirements. Although it can be resolved by refinement, this interesting issue reflects models' strategies when handling conflicts between training data patterns and task instructions.

**Crash Analysis.** Table 4 presents the crash analysis results from fuzzing LLM-generated apps (full version available in the Appendix C). First, the "evade development" strategies employed by GPT-4.1 ultimately backfire at runtime. While achieving higher compilation rates, the app fundamentally fails to start when executed, indicating that evasive compilation error fixes often intro-

Table 4: Runtime crash analysis across LLMs.

| Model | Native Crash | | Failed to Start | |
|---|---|---|---|---|
| | w/o Fix | w/ Fix | w/o Fix | w/ Fix |
| GPT-4.1 | 0.0 | 11.0 | 2.0 | **66.0** |
| Claude-Opus | **48.0** | **48.0** | 9.0 | 11.0 |
| Gemini-Pro | 25.0 | 37.0 | 14.0 | 21.0 |
| GPT-5-High | 21.0 | 0.0 | 5.0 | 25.0 |

duce fundamental flaws, such as incomplete resource initialization that prevent proper app bootstrapping. Second, notably all crashes are native crashes rather than Java-level exceptions, indicating that the generated Java code itself is generally robust with proper exception handling. This result suggests that existing LLMs excel at defensive programming practices and maintain good exception handling patterns as illustrated in Figure 7. However, crashes occur when calling third-party libraries or interacting with OS services due to parameter validation failures and contract mismatches. While LLMs demonstrate solid understanding of Java code, they lack sufficient knowledge of underlying implementation details and resource constraints. Consequently, seemingly safe Java code can trigger native-level issues when interfacing with lower-level components, highlighting the gap between surface-level language proficiency and deep system understanding required for software engineering.

## 5 CONCLUSION

In this paper, we have introduced APPFORGE, a comprehensive benchmark for evaluating LLMs on real-world Android app development from scratch, revealing significant gaps between current capabilities and practical software engineering requirements. Through systematic evaluations of 12 state-of-the-art LLMs across 101 diverse development scenarios, we have found that even the best-performing models achieve only modest success rates, contrasting sharply with their high performance on existing code generation benchmarks, suggesting that fundamental innovations rather than incremental improvements may be necessary toward fully automated software engineering.

## ACKNOWLEDGMENTS

We thank the Android development and testing experts at WeChat, Tencent, for their kind and voluntary help in reviewing the benchmark. This work was supported in part by the Fundamental and Interdisciplinary Disciplines Breakthrough Plan of the Ministry of Education of China (No. JYB2025XDXM118). The work by Dezhi Ran and Tao Xie is partially supported by the National Natural Science Foundation of China under Grant Nos. 623B2006, 92464301, and U25A6023. Any opinions, findings, conclusions, or recommendations expressed in this material are those of the authors and do not necessarily reflect the views of the funding agencies.

## ETHICS AND REPRODUCIBILITY STATEMENT

This work adheres to the ICLR Code of Ethics. No human subjects or animal experimentation were involved. No personally identifiable information was used, and no experiments posed privacy or security risks. We are committed to transparency and integrity throughout the research process. We believe that APPFORGE can be used for various purposes, including evaluating the cutting-edge capabilities of code LLMs for software engineering, training better software engineering models and agents, and as a seed benchmark for building larger benchmarks for app development. We have strictly adhered to the license of open-source apps since we use the runtime behavior of these apps instead of using their source code for constructing APPFORGE. However, we are concerned about potential misuse of APPFORGE for training models to reverse engineering existing Android apps, making the plagiarism of apps a practical concern.

As a benchmark paper, the benchmark has been made publicly available at `https://github.com/TongmingLAIC/AppForge`, providing detailed documentation, leaderboard, and dockerized environment to ensure easy reproduction and customized use. We have detailed the selection criteria in our Appendix and used popular open-source apps in F-Droid to allow reproducibility of task collection. The experimental setup is described in detail.

## LLM USAGE

LLMs were employed solely to assist in writing and polishing the manuscript, including refining language, improving readability, and enhancing clarity. The LLM was used for tasks such as sentence rephrasing, grammar checking, and improving overall flow.

The LLM was not involved in ideation, research methodology, or experimental design. All research concepts, analyses, and results were developed and conducted by the authors. The authors take full responsibility for the manuscript content, including any text generated or polished by the LLM, and confirm that all LLM-assisted text adheres to ethical guidelines and does not constitute plagiarism or scientific misconduct.

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

APPENDIX

## A  DETAILED DISCUSSION OF RELATED WORK

**Machine Learning for Software Engineering.** Machine learning, including large language models (LLMs), is increasingly used to address real-world software engineering tasks due to their advantages over traditional program analysis techniques. Typical use cases include automatic code generation (Chen et al., 2021; Austin et al., 2021; Liu et al., 2023; Chen et al., 2024a), malware detection (Qian et al., 2025; Sahs & Khan, 2012; Chen et al., 2020), test generation (Chen et al., 2024b; Ryan et al., 2024; Schäfer et al., 2023), and program repair (Jimenez et al., 2024; Jin et al., 2023; Xia & Zhang, 2024; Yang et al., 2024).

Most relevant to our APPFORGE are approaches that apply LLMs to automated code generation or code completion (Chen et al., 2025; Wang et al., 2023; Athiwaratkun et al., 2023). However, existing code generation datasets are largely limited to the function level (Chen et al., 2021; Austin et al., 2021; Liu et al., 2023; Chen et al., 2025; Yu et al., 2024), and repository-level work focuses mainly on code completion rather than generation from scratch (Liu et al., 2024; Bairi et al., 2024). Compared with existing datasets, APPFORGE introduces a more realistic and challenging setting for evaluating the capability of LLMs to perform software development from scratch. This setting better reflects real-world development scenarios where models must synthesize coherent, functional, and maintainable codebases instead of completing the missing lines in existing codebases.

**Code Generation Benchmarks.** Many benchmarks have been proposed to evaluate the code generation capabilities of LLMs (Guan et al., 2025; Chen et al., 2024a; Yu et al., 2024; Jimenez et al., 2024; Mathai et al., 2024). HumanEval and MBPP introduce human-crafted datasets focused on synthesizing function-level code from natural language descriptions and have become standard benchmarks (Chen et al., 2021; Austin et al., 2021). Building on this work, HumanEval-XL (Peng et al., 2024) extends HumanEval to support multilingual settings. Moreover, EvalPlus (Liu et al., 2023) highlights limitations in HumanEval and MBPP, particularly their limited test case coverage, and proposes a more rigorous evaluation benchmark. BigCodeBench (Zhuo et al., 2024) introduces a larger-scale benchmark designed to further evaluate LLMs' code generation capabilities. Beyond these function-level code generation benchmarks, recent repository-level benchmarks have also been proposed. For example, SWE-Bench (Jimenez et al., 2024) focuses on evaluating LLMs' patch generation ability at the repository level. Despite effective, most benchmarks are static and lag behind LLM advancements, prompting the emergence of dynamic benchmarks for up-to-date, contamination-free evaluation. LiveCodeBench (Jain et al., 2024) collects newly released programming completion problems from online coding platforms to minimize data contamination. PPM (Chen et al., 2024a) and DyCodeEval (Chen et al., 2025) propose an automated approach to generate new benchmark data at the evaluation stage, mitigating potential data contamination. SWE-Bench-Live (Zhang et al., 2025) follows the LiveCodeBench schema to collect new patches from GitHub repositories, providing a continuous and realistic evaluation environment.

Compared to existing code generation benchmarks, APPFORGE operates at the repository level and includes rigorous evaluation. It can be dynamically constructed by collecting latest projects from F-Droid, preserving a much broader range of challenges rooted in real-world software development—going beyond closed-form completion or patch generation.

**Android App Ecosystem.** Android apps represent one of the most significant software ecosystems globally, with over 2.6 million apps available on Google Play Store and millions more distributed through alternative channels (Technource, 2022). In Android ecosystems, F-Droid (2025c) is the leading open-source repository, serving millions of users worldwide. Its rigorous review process and anti-feature labeling ensure high-quality software that often exemplifies Android development best practices. The repository spans diverse categories—productivity, multimedia, utilities, games, and specialized professional tools—many maintained by experienced developers and organizations (2025a; 2025b; 2025).

Android apps, typically built with Java or Kotlin following Material Design and Android architecture guidelines, comprise multiple interconnected components such as Activities, Fragments, XML layouts, and manifest configurations (2025a). Developing robust apps demands expertise across UI design, background services, data persistence, networking, device optimization, and security (Mayrhofer et al., 2021; Developers, 2025b; GeeksforGeeks, 2025; Developers, 2025c). Given the prevalence and

complexity of the Android ecosystem, developing Android apps automatically provides significant commercial and practical impact. APPFORGE, filling a critical gap, provides the first benchmark for assessing LLM capabilities in Android development.

# B DETAILS OF APPFORGE

## B.1 TASK INSTANCES

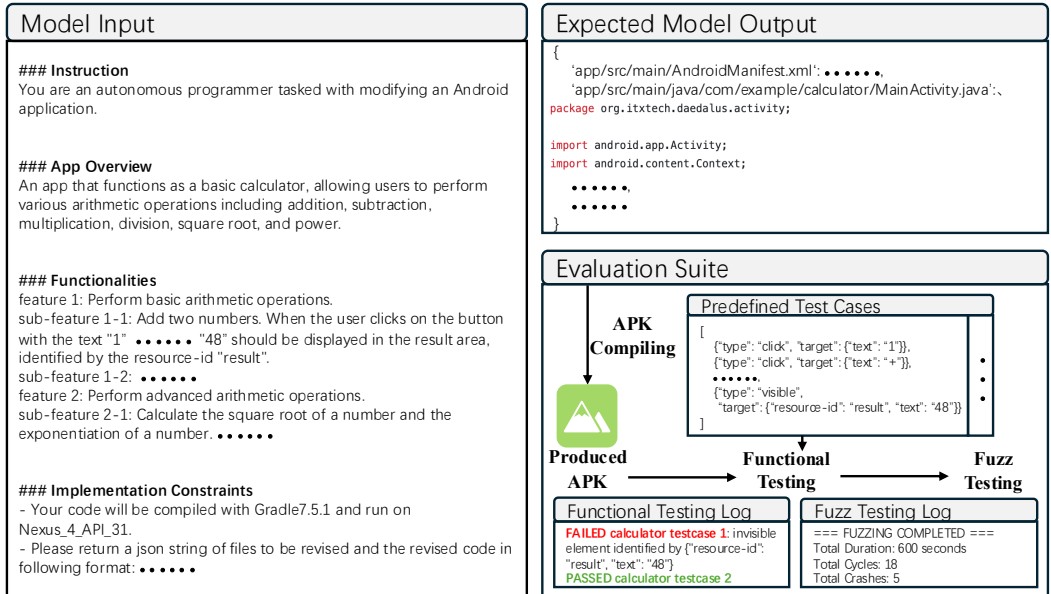

Figure 11: Example of Task Instance.

## B.2 STATISTICS OF SEED APPS.

Table 5: Statistics of Seed Apps used for Constructing APPFORGE.

|         | # LoC    | # Acts | # Files |
|---------|----------|--------|---------|
| **Range**  | 194–53K  | 1–21   | 2–508   |
| **Avg.**   | 6367.2   | 3.7    | 44.0    |
| **Median** | 4530     | 3      | 26      |

## B.3 EVALUATION METRIC CALCULATION

We employ four primary metrics that directly correspond to the three evaluation stages to comprehensively assess the quality and functionality of generated Android code.

The **compilation rate** (Eq. 1) measures the percentage of generated code that successfully compiles into valid APKs without syntax or dependency errors during the compilation stage, indicating the basic correctness and completeness of the generated code structure. Here, $N_{\text{compiled}}$ denotes the number of successfully compiled cases and $N_{\text{total}}$ denotes the total number of tasks.

The **test pass rate** (Eq. 2) evaluates, using *macro averaging*, the mean percentage of predefined test cases that pass across all compiled apps during the testing stage. This metric reflects how well the generated code implements the specified functionalities and meets behavioral requirements. For each compiled app $i$, $t_{\text{passed}}^{(i)}$ denotes the number of passed test cases, and $t_{\text{total}}^{(i)}$ denotes the total number of

test cases for that app. The macro average is computed by taking the mean of per-app pass rates over all $N_{\text{compiled}}$ compiled apps.

The **crash rate** (Eq. 3) quantifies the percentage of compiled apps that crash or terminate unexpectedly during the fuzz testing stage, assessing the robustness and stability of the generated code under various stress scenarios. Here, $N_{\text{crashed\_apps}}$ denotes the number of compiled apps that experienced crashes during fuzz testing, and $N_{\text{compiled}}$ denotes the total number of compiled apps.

Additionally, the **functional success rate** (Eq. 4) measures the percentage of tasks that achieve both successful compilation and complete test suite passage, representing the overall functional correctness of the generated apps regardless of their robustness under stress testing. Here, $N_{\text{compiled\_and\_tests\_passed}}$ denotes the number of tasks meeting both criteria.

The pipeline captures detailed logs and execution traces throughout all three stages, enabling precise computation of these metrics and comprehensive result reporting for large-scale benchmarking experiments.

$$\text{Compilation Rate} = \frac{N_{\text{compiled}}}{N_{\text{total}}} \times 100\% \tag{1}$$

$$\text{Test Pass Rate} = \frac{1}{N_{\text{compiled}}} \sum_{i=1}^{N_{\text{compiled}}} \left( \frac{t_{\text{passed}}^{(i)}}{t_{\text{total}}^{(i)}} \right) \times 100\% \tag{2}$$

$$\text{Crash Rate} = \frac{N_{\text{crashed\_apps}}}{N_{\text{compiled}}} \times 100\% \tag{3}$$

$$\text{Functional Success Rate} = \frac{N_{\text{compiled\_and\_tests\_passed}}}{N_{\text{total}}} \times 100\% \tag{4}$$

### B.4 IMPLEMENTATION DETAILS OF APPFORGE

To ensure consistent and reproducible evaluation across different systems, we establish a standardized execution environment that supports the entire evaluation workflow described above. The environment consists of two main components that work together to enable seamless automated evaluation. First, we utilize the official Android Emulator with API level 31 (Android 12) running on x86_64 architecture for APK installation and testing. This emulator configuration is packaged into a Docker container that guarantees seamless deployment on Ubuntu systems without requiring any additional manual configuration. The containerized approach eliminates environment-specific issues and ensures that all evaluations are conducted under identical conditions, enabling reliable APK execution and test case validation. Second, we configure a standardized Gradle build environment within the same container to handle the automated compilation process. This environment includes pre-installed Android SDK components, build tools, and dependency management configurations that are commonly used in Android development. The Gradle setup is optimized for automated compilation of generated code files into APKs, with appropriate timeout settings and resource allocation to handle various code complexity levels. The build environment is configured to provide detailed compilation error messages when needed, supporting the iterative refinement workflow for models that can benefit from error feedback.

### B.5 RANKING CRITERIA OF APPS IN F-DROID

The ranking of F-Droid apps follows a sequential process that begins with filtering the dataset to retain only actively maintained and popular apps, requiring at least 50 GitHub stars, 10 forks, non-archived status, an update date no earlier than 2022, and primary implementation in Java or Kotlin. For each app that passes the filter, an LLM assigns integer scores from 1 to 5 for four evaluation aspects: maintainability, which measures how easy the source code is to understand and modify; reproducibility, which reflects the ability to produce consistent results under stable conditions; generality, which indicates how broadly the app can operate across different devices and configurations; and evaluation efficiency, which assesses the speed and resource requirements of building and testing the app. The average of these four scores forms the quality rating. The difficulty ranking is then determined by combining the LLM-generated complexity score with the number of

activities in the app and the total code size in bytes. Based on these combined criteria, the ranking process selects the top 40 apps within each difficulty level, resulting in a final list of 200 top-ranked apps that are representative, maintainable, reproducible, and diverse in technical challenge, ensuring a comprehensive benchmark for assessing the Android development capabilities of LLMs.

### B.6   PROMPT EXAMPLES

```
1  You are an autonomous programmer and you are modifying a default
       Android app template with empty activity.
2  Your code will be compiled with Gradle7.5.1 and run on
       Nexus_4_API_31.
3  The default Android app template "File Structure" is shown below.
       You can replace or add some files in the templates to
       implement the app.
4  "File Structure":
5  |-- app
6  | |-- build.gradle
7  | --- src
8  | |-- main
9  | | |-- AndroidManifest.xml
10 ...
11
12 Your app should implement every feature in "App Features", and we'
       ll test on each of the features. Note that you should pay
       attention to the resource-id, content-desc, texts and other
       attributes we provide with corresponding widgets in "App
       Features" and exactly match the attributes when implementing
       the widgets.
13 "App Features":
14 description: An Android app for quickly sharing your current
       location.
15 feature 1: Share your location via ...
16 ...
17
18 Please return a json string and only a json string of files to be
       revised and the revised code in following format:
19 {
20     "app/src/main/AndroidManifest.xml":...,
21     ...
22 }
```

## C   DETAILED EXPERIMENT RESULTS

### C.1   SETUP

All LLMs use identical task prompts provided by APPFORGE as well as the same hyperparameter settings(with the temperature set to 0.2 using greedy decoding (Brown et al., 2020)).

### C.2   DETAILED STATISTICS

Figures 6 and  7 present the detailed statistics of compilation errors and runtime errors of the apps developed by different LLMs.

### C.3   FULL EXAMPLE OF DEFENSIVE PROGRAMMING BY GPT-5.

Listing 1 presents the full example of the defensive programming achieved by GPT-5.

Table 6: Compilation results of LLMs.

| Model | Compile Success | Package Name not Found | Android Resource Linking Failed | Resource Compilation Failed | Wrong JSON Format | Uses / Overrides Deprecated API | Cannot Find Symbol | Exported Flag Missing | Requires CompileSdkVersion | Compilation Timeout or Unstable | Others |
|---|---|---|---|---|---|---|---|---|---|---|---|
| *Pass@1* | | | | | | | | | | | |
| Claude-4-Sonnet | 41.0 | 8.0 | 34.0 | 1.0 | 0.0 | 0.5 | 2.0 | 4.0 | 5.0 | 0.0 | 5.5 |
| Claude-5-Opus | 81.0 | 0.0 | 4.0 | 2.0 | 0.0 | 2.92 | 5.17 | 0.0 | 1.0 | 1.0 | 3.92 |
| Gemini-2.5-Pro | 54.0 | 0.0 | 14.0 | 0.1 | 4.0 | 0.67 | 5.83 | 0.0 | 6.9 | 7.0 | 8.5 |
| GPT-4.1 | 7.0 | 0.0 | 6.0 | 7.33 | 0.0 | 0.06 | 3.94 | 72.67 | 0.0 | 0.0 | 4.0 |
| GPT-5-High | 46.0 | 0.0 | 7.0 | 1.63 | 4.0 | 3.5 | 6.5 | 11.67 | 18.71 | 0.0 | 2.0 |
| Qwen3-Coder | 28.0 | 0.0 | 52.0 | 1.0 | 4.0 | 0.0 | 7.0 | 4.0 | 0.0 | 0.0 | 5.0 |
| GLM-4.5 | 25.0 | 4.0 | 31.0 | 6.11 | 3.0 | 3.48 | 13.63 | 2.0 | 1.89 | 1.0 | 9.89 |
| DeepSeek-R1 | 15.0 | 4.0 | 62.0 | 2.09 | 2.0 | 0.83 | 3.17 | 5.0 | 0.0 | 0.0 | 6.91 |
| DeepSeek-V3 | 6.0 | 27.0 | 28.0 | 1.0 | 35.0 | 0.0 | 0.0 | 1.0 | 0.0 | 0.0 | 3.0 |
| Kimi K2 | 17.0 | 5.0 | 36.0 | 4.67 | 2.0 | 0.93 | 8.07 | 10.33 | 9.0 | 0.0 | 8.0 |
| *with Compilation Error Feedback* | | | | | | | | | | | |
| Claude-4-Sonnet | 78.0 | 1.0 | 17.0 | 0.0 | 0.0 | 0.44 | 2.56 | 0.0 | 0.0 | 0.0 | 2.0 |
| Claude-5-Opus | 91.0 | 0.0 | 2.0 | 1.0 | 0.0 | 0.24 | 5.76 | 0.0 | 0.0 | 1.0 | 0.0 |
| Gemini-2.5-Pro | 69.0 | 0.0 | 8.0 | 0.0 | 0.0 | 0.42 | 8.58 | 0.0 | 2.0 | 5.0 | 8.0 |
| GPT-4.1 | 75.0 | 0.0 | 13.0 | 1.93 | 0.0 | 0.3 | 6.7 | 4.07 | 0.0 | 0.0 | 0.0 |
| GPT-5-High | 83.0 | 0.0 | 6.0 | 2.0 | 0.0 | 0.0 | 4.0 | 5.0 | 0.9 | 0.0 | 0.1 |
| Qwen3-Coder | 86.0 | 0.0 | 7.0 | 0.0 | 0.0 | 0.58 | 3.5 | 0.0 | 0.0 | 1.0 | 2.92 |
| GLM-4.5 | 45.0 | 3.0 | 29.0 | 3.0 | 2.0 | 0.32 | 9.68 | 0.0 | 1.0 | 2.0 | 6.0 |
| DeepSeek-R1 | 45.0 | 9.0 | 25.0 | 1.0 | 0.0 | 1.63 | 12.37 | 2.0 | 1.0 | 0.0 | 4.0 |
| DeepSeek-V3 | 27.0 | 8.0 | 26.0 | 2.0 | 31.0 | 0.24 | 1.0 | 3.0 | 0.0 | 0.0 | 2.76 |
| Kimi K2 | 42.0 | 2.0 | 22.0 | 2.0 | 2.0 | 0.43 | 12.57 | 3.0 | 8.0 | 0.0 | 7.0 |

Table 7: Application Runtime Error Statistics

| Model | Pass@1 | | | with Compilation Error Feedback | | |
|---|---|---|---|---|---|---|
| | ANR | Native Crash | Failed Start | ANR | Native Crash | Failed Start |
| Kimi K2 | 0.0 | 11.0 | 3.0 | 1.0 | 13.0 | 18.0 |
| DeepSeek-V3 | 0.0 | 4.0 | 2.0 | 0.0 | 11.0 | 4.0 |
| Qwen3-Coder | 0.0 | 19.0 | 4.0 | 0.0 | 16.0 | 9.0 |
| GLM-4.5 | 0.0 | 15.0 | 5.0 | 0.0 | 27.0 | 15.0 |
| DeepSeek-R1 | 0.0 | 8.0 | 4.0 | 1.0 | 25.0 | 6.0 |
| GPT-4.1 | 0.0 | 0.0 | 2.0 | 0.0 | 11.0 | 66.0 |
| Claude-5-Opus | 0.0 | 48.0 | 9.0 | 0.0 | 48.0 | 11.0 |
| GPT-5-High | 1.0 | 21.0 | 5.0 | 0.0 | 0.0 | 25.0 |
| Gemini-2.5-Pro | 0.0 | 25.0 | 14.0 | 0.0 | 37.0 | 21.0 |
| Claude-4-Sonnet | 0.0 | 21.0 | 10.0 | 0.0 | 0.0 | 20.0 |

Listing 1: Case study showing proactive defensive programming implementation by GPT-5 in the Autostarts app, including robust null-safety checks, multi-level fallback strategies for system navigation, and guarded UI state transitions beyond basic requirements.

```java
// --- Null-safety on user input (avoid NPE) ---
private void filterList(String query) {
  String q = (query == null) ? "" : query.trim().toLowerCase(Locale.ROOT);  // defensive
  for (AppEntry e : appEntries) {
    boolean match = q.isEmpty() || e.title.toLowerCase(Locale.ROOT).contains(q);
    e.row.setVisibility(match ? View.VISIBLE : View.GONE);
  }
}

// --- UI state guard: close search safely and reset filters ---
private void closeSearchIfOpen() {
  if (searchContainer.getVisibility() == View.VISIBLE) {  // defensive visibility check
    searchContainer.setVisibility(View.GONE);
    searchInput.setText("");                              // reset state
    filterList("");                                       // restore full list
  }
}

@Override
public void onBackPressed() {
  if (searchContainer != null && searchContainer.getVisibility() == View.VISIBLE) { // guard
    closeSearchIfOpen();   // consume back to close search instead of leaving activity
    return;
  }
  super.onBackPressed();
}

// --- Best-effort navigation with graceful degradation ---
private void openAppInfo(String packageName) {
  try {
    Intent it = new Intent(Settings.ACTION_APPLICATION_DETAILS_SETTINGS);
    it.setData(Uri.parse("package:" + packageName));
    it.addFlags(Intent.FLAG_ACTIVITY_NEW_TASK);
```

```
34      startActivity(it);
35    } catch (ActivityNotFoundException e) {       // device/ROM mismatch fallback
36      try {
37        Intent fallback = new Intent(Settings.ACTION_APPLICATION_DETAILS_SETTINGS);
38        fallback.setData(Uri.parse("package:" + getPackageName()));  // fallback to self
39        startActivity(fallback);
40      } catch (Exception ex) {                     // last resort: user-visible error
41        Toast.makeText(this, "Unable to open Application Info", Toast.LENGTH_SHORT).show();
42      }
43    }
44 }
45
46 // --- Defensive null-check on optional view ---
47 private void showMenuForGoogleDuring() {
48    new AlertDialog.Builder(this)
49      .setTitle("Google")
50      .setItems(new CharSequence[]{"Disable", "Application Info"}, (d, which) -> {
51        if (which == 0) {
52          if (statusGoogleDuring != null) {       // view presence not guaranteed across
        layouts
53            statusGoogleDuring.setVisibility(View.VISIBLE);
54          }
55          Toast.makeText(this, "Google autostart disabled for During Startup", Toast.
        LENGTH_SHORT).show();
56        } else if (which == 1) {
57          openAppInfo("com.google.android.googlequicksearchbox");
58        }
59      })
60      .show();
61 }
```

