# OpenReview forum: "From Assistant to Independent Developer — Are GPTs Ready for Software Development?"
_ICLR.cc/2026/Conference — ICLR 2026 Poster_

### Official Review · Reviewer_WwYL · 2025-10-25

**Soundness:** 3
**Presentation:** 3
**Contribution:** 2
**Rating:** 4
**Confidence:** 3

**Summary:**

The paper presents a benchmark called AppForge. The goal is to evaluate LLM's capability, beyond generating isolated snippets or functions. The argument is that existing benchmarks do not test system-level reasoning and integration required to build a complete application. To this end, AppForge builds on Android application development as the benchmark domain, and it is based on the top 200 highest-scoring apps across different categories. Evaluations compare how 12 LLMs perform on AppForge.

**Strengths:**

Reading this paper brings back the fond memory of developing Android apps.

1. I can see how app development can support the claim of going beyond generating isolated code snippets and functions. The idea of using Android apps as the benchmark domain seems reasonable.

2. While I haven't personally used a lot of LLM benchmarks, the explanation on how AppForge is different from others is clear to me.

3. I also like that AppForge includes a diverse range of apps. And, evaluations (over 12 LLMs) suggest that AppForge can differentiate model capabilities better than existing code generation benchmarks.

**Weaknesses:**

I have been thinking whether I would use AppForge myself, and the following aspects are not clear to me:

1. What LLM capabilities is AppForge really testing? I understand that the end-goal is generating fully-working Andorid apps, but app development is simply a scenario, not a capability. From the introduction, the capability seems to be system-level reasoning, but what is system-level reasoning? As a system researcher, I find that system-level reasoning can be an overloaded term: reasoning about system behavior, properties, states, and so on?

2. Is the scoring system just pass-or-fail? Developing a fully-working app is non-trivial, and it involves a lot of planning and implementation steps. Is it possible that minor mistakes along the way result in app failure?

3. AppForge is based on top 200 highest-scoring apps across many categories. I am wondering whether solution leaking would be a problem. In other words, are there open-sourced version of these 200 apps, whose source codes are present in the LLM training data? After all, these are highest-scoring apps, and it is likely that people have released open-sourced version?

4. One minor question. Software development is typically imposed with budget constraints. How does AppForge ensure that the better-performing LLMs are not using excessively more budgets than others?

**Questions:**

Please refer to the "Weaknesses" section, for my questions.

---

> ### Author Response · Authors · 2025-11-21
> **Response (1/2)**
>
> We thank the reviewer for the insightful and constructive comments. We are especially grateful for the appreciation that AppForge goes beyond isolated code generation, being different from other benchmarks, and differentiates model capabilities effectively. We address key questions and concerns as below. Please kindly let us know if you have further questions or concerns.
>
> > ​*W1*: **What LLM capabilities is really tested by AppForge:** From the introduction, the capability seems to be system-level reasoning, but what is system-level reasoning? System-level reasoning can be an overloaded term: reasoning about system behavior, properties, states, and so on?
>
> Sorry for the confusion. AppForge is designed to evaluate an LLM’s **systematic generative synthesis** capability. Specifically, its ability to perform **architectural planning**, **solution design**, and **from-scratch construction** of full applications. In our setting, the model must carry out **zero-context code generation**, synthesizing long, coherent application code directly from high-level requirements, runtime constraints, and self-generated contextual assumptions. Unlike infilling-style code generation, this capability demands constructing a complete solution from scratch instead of merely extending an existing code fragment. We will add the above definition and explanation clearly in revision.
>
> > ​*W2*: **scoring system:** Is the scoring system just pass-or-fail? Developing a fully-working app is non-trivial, and it involves a lot of planning and implementation steps. Is it possible that minor mistakes along the way result in app failure?
>
> Thank you for this important question about our evaluation granularity. We want to clarify that AppForge does not use a simple binary pass-or-fail system. Our evaluation framework includes multiple metrics that discriminate between different levels of completeness and functionality.
>
> Our evaluation pipeline measures four distinct dimensions. First, compilation success captures whether the generated code can be built without errors, which tests basic syntax correctness and dependency management. Second, functional correctness is measured by the percentage of test cases passed, ranging from 0% (completely non-functional) to 100% (all features working). Third, robustness is evaluated through automated fuzzing that simulates edge cases and user errors, measuring crash rates and error handling. These metrics allow us to distinguish between apps that completely fail to compile, apps that compile but crash immediately, apps with partial functionality, and fully working apps.
>
> Additionally, AppForge tasks naturally vary in complexity, which self-contain different numbers of planning and implementation steps, ranging from relatively simple single-Activity apps to highly complex apps requiring sophisticated state management, background services, and complex user interactions. This diversity means that models demonstrating competence on simpler tasks can still receive meaningful scores even if they fail on the most complex ones. The distribution of task difficulties provides a natural gradient for evaluation rather than a single pass-fail threshold.
>
> We also want to address the philosophical question about expectations for AI systems. Modern LLM-based coding assistants have evolved significantly, and the community's expectations have shifted accordingly. Following Anthropic and OpenAI's definition of "independent working hours" [1,2], we believe current frontier models should be capable of completing tasks that involve substantial planning steps and self-correcting minor mistakes without human intervention. AppForge is designed to evaluate whether models have reached this threshold of autonomy for application-level development. A fully-working app is indeed non-trivial, but this is precisely the capability we aim to measure. If minor mistakes consistently result in app failure, this indicates that current models have not yet achieved the level of robustness and self-correction needed for autonomous development.
>
> For researchers interested in evaluating models with greater tolerance for minor mistakes or measuring incremental progress on smaller code units, existing benchmarks such as HumanEval [3], MBPP [4], and ClassEval [5] are suitable. These benchmarks focus on function-level or class-level code generation, where partial correctness is easier to define and measure. AppForge complements these benchmarks by targeting the higher bar of system-level software engineering, where integration, coordination, and end-to-end functionality are essential. Our results showing that even the best models achieve only 18.8% success rate suggest that this capability gap, building complete and working applications, remains a significant frontier for AI-assisted software development.

---

> ### Author Response · Authors · 2025-11-21
> **Response (2/2)**
>
> *[1] Anthropic. (2025, September 29). Introducing Claude Sonnet 4.5. https://www.anthropic.com/news/claude-sonnet-4-5*
>
> *[2] OpenAI. (2025, November 19). Building more with GPT-5.1-Codex-Max. https://openai.com/index/gpt-5-1-codex-max/#long-running-tasks*
>
> *[3] M. Chen et al., Evaluating large language models trained on code, arXiv preprint arXiv:2107.03374, 2021.*
>
> *[4] J. Austin et al., Program synthesis with large language models, arXiv preprint arXiv:2108.07732, 2021.*
>
> *[5] X. Du et al., ClassEval: A manually-crafted benchmark for evaluating LLMs on class-level code generation, arXiv preprint arXiv:2308.01861, 2023.*
>
> > *W3*: **Solution leaking:** I am wondering whether solution leaking would be a problem. In other words, are there open-sourced version of these 200 apps, whose source codes are present in the LLM training data? After all, these are highest-scoring apps, and it is likely that people have released open-sourced version?
>
> Thanks for your excellent question! Solution leaking is a critical problem in constructing benchmarks for LLMs and we have tried our best to mitigate the solution leaking issue for two aspects. First, we use the runtime behavior instead of code implementation of F-droid apps to construct AppForge. Second, we deliberately transform the resource identifiers in task descriptions and test cases to make the task different from the F-droid apps’ original implementation while keeping task authenticity.
>
> To further investigate to what extent the solution leaking exists, we conduct a code similarity analysis between the code generated by LLMs and the code written in the F-droid. The following table presents the results of our code similarity analysis using JPlag [6], with a breakdown of the average and maximum scores for each model across tasks in AppForge.
>
> | Model Name | Average Similarity | Max Similarity |
> |:----------------|:-------------------|:---------------|
> | Claude-4-opus | 0.058 | 0.226 |
> | DeepSeek-V3 | 0.047 | 0.178 |
> | Gemini-2.5-pro | 0.045 | 0.219 |
> | GPT-5-high | 0.050 | 0.178 |
> | Qwen3-coder | 0.038 | 0.195 |
>
> The findings indicate that the degree of code similarity between original F-droid app implementation and LLM-generated implementation is minimal. The average score across all outputs is approximately 0.05, with the highest recorded similarity not exceeding 0.23. More granularly, 99.5% of the generated outputs fall below a 20% similarity threshold, while 91% are below a 10% threshold. To validate these quantitative results, we inspect the top 10 samples exhibiting the highest similarity, which found no significant evidence of solution leaking except for common code patterns in Android development.
> We will add the detailed similarity analysis and discuss the solution leaking in our appendix D in our revision.
>
> *[6] JPlag. (n.d.). JPlag - Detecting Software Plagiarism. GitHub. https://github.com/jplag/JPlag*
>
>
> > ​*W4*: **Budget constraints:** How does AppForge ensure that the better-performing LLMs are not using excessively more budgets than others?
>
> We fully agree that budget constraint is a practical concern in real-world software development. While our main paper focused on performance of LLMs, we have logged the costs of different LLMs measured by the official token prices of their API services shown in the table below.
>
> | Model | Cost  |
> |------|-------------|
> | Claude 4 Opus | $49.2646 |
> | Claude 4 Sonnet | $10.6149 |
> | DeepSeek R1 | $0.5840 |
> | DeepSeek V3 | $0.4208 |
> | Gemini 2.5 Pro | $6.6452 |
> | GLM-4.5 | $1.3476 |
> | GPT-4.1 | $4.2597 |
> | GPT-5-high | $6.3741 |
>
> In general, the proprietary LLMs achieve better performance at higher costs.
> Notably, while Claude-4-Opus achieves a higher success rate than Gemini-2.5-Pro, it costs seven times more than Gemini-2.5-Pro. When taking the budget constraint into consideration, Gemini-2.5-pro could be a better choice than Claude.  We will add the cost analysis in Appendix C.4 in our revision.

---

### Official Review · Reviewer_TWQk · 2025-10-25

**Soundness:** 3
**Presentation:** 3
**Contribution:** 3
**Rating:** 6
**Confidence:** 4

**Summary:**

This paper presents APPFORGE, a novel benchmark designed to evaluate the end-to-end software development capabilities of large language models (LLMs) in the context of Android app development. Unlike existing benchmarks that primarily assess function-level or patch-level code generation, APPFORGE targets real-world software engineering tasks, including system-level reasoning, component integration, UI design, state management, and exception handling. The benchmark includes 101 tasks automatically constructed and validated from real-world open-source Android apps, and features a fully automated evaluation pipeline involving compilation, test execution, and fuzzing. The authors evaluate 12 state-of-the-art LLMs (including GPT-5, Claude-Opus, and others), revealing that current models perform poorly on this benchmark—with a maximum success rate of only 18.8%—thus highlighting the significant gap between current capabilities and the demands of real-world software development.

**Strengths:**

1. The first to comprehensively evaluate LLMs in full Android application development from scratch, addressing a crucial gap in current code evaluation benchmarks.
2. Tasks are derived from real-world, actively maintained F-Droid apps, ensuring diversity and authenticity.
3. The benchmark combines automatic task construction and evaluation with expert validation, ensuring high-quality and reproducible results.
4. The paper provides a breakdown of LLM failure cases, behavioral patterns (e.g., avoidance strategies), and complexity trends.

**Weaknesses:**

While the paper identifies that some models (e.g., GPT-4.1) evade development by deleting faulty code instead of fixing it, it does not offer a detailed exploration of this behavior. There is no case analysis or inspection of prompts provided to the model during iterations. The paper does not clarify whether this is due to training data bias, a form of “lazy” behavior by the model, or a failure in prompt engineering. This limits the explanatory depth of an otherwise critical observation.

The title poses a broad question—“Are GPTs Ready for Software Development?”—but the empirical scope is restricted to Android app development. While Android development is non-trivial, its characteristics (e.g., strict lifecycle constraints, XML UI design, framework conventions) do not generalize to other domains such as backend systems, web applications, or embedded systems. The gap between the title’s generality and the study’s actual scope should be addressed.

**Questions:**

1. In Table 1, are the results under “with Compilation Error Feedback” obtained from a single iteration or multiple rounds of refinement? For instance, Qwen3-Coder reports 85.15% compilation success, but this seems inconsistent with the values in Figure 5. In Figure 5, on the line for Qwen3-Coder Compile, there seem to be no node values corresponding to 85.15%. Please clarify.
2. Could the authors provide the prompt used for iterative generation based on the previous compilation error output?
3. For the observed “avoidance behavior”, could the authors provide concrete cases and detailed analysis? Specifically, is this phenomenon a result of training data bias, a form of lazy model strategy, or prompt design shortcomings?

---

> ### Author Response · Authors · 2025-11-21
> **Response (1/2)**
>
> We sincerely thank you for the insightful and constructive feedback. We address key questions and concerns as below. Please kindly let us know if you have further questions or concerns.
>
>
> > ​*W1 & Q3*: **concrete cases and detailed analysis of avoidance behavior:** There is no case analysis or inspection of prompts provided to the model during iterations. The paper does not clarify whether this is due to training data bias, a form of “lazy” behavior by the model, or a failure in prompt engineering. This limits the explanatory depth of an otherwise critical observation. Could the authors provide concrete cases and detailed analysis? Specifically, is this phenomenon a result of training data bias, a form of lazy model strategy, or prompt design shortcomings?
>
> Following your suggestion, we present the case analysis and inspection of prompts as below. Specifically, our initial prompt shown in Appendix B.6 does not focus on asking model to implement every function (which we considered a requirement that does not need to be explicitly stated), which might lead to the avoidance behavior where GPT4.1 and Kimi K2 evade development tasks during iterative refinement by deleting faulty implementations instead of repairing them. To study the causes, we change the prompt to several new versions, such as adding
> ```
> Generate complete code where every function is fully implemented. Avoid:
> - Placeholder comments like "// implement this"
> - Empty function bodies
> - "TODO" statements
> - Function stubs without actual code
> Ensure all functions contain their actual working implementation.
> ```
> and other prompt variations. These versions emphasize the importance of maintaining functionality. These prompts do take effect and the experimental statistics show that the avoidance behavior disappears. For Kimi K2, the average #LOC becomes 312.65 after iterative refinement with the new prompt.
>
> However, since all models use the same prompt in our evaluation and other models do not exhibit the avoidance behavior, we believe the avoidance behavior is a model-specific characteristic of GPT4.1 and Kimi K2, such as aggressive reward optimization for compilation success rate when more than two and layered rewards are given. Given the limited availability of training data and details in GPT4.1 and Kimi K2, we leave further investigation as future work, where we plan to conduct more fine-grained analysis on truly open models like OLMo and StarCoder.
>
> We will add the concrete case analyses and more discussion to Appendix C.5 in our revision.
>
>
> > ​*W2*: **gap between the title’s generality and the study’s actual scope:** The title poses a broad question—“Are GPTs Ready for Software Development?”—but the empirical scope is restricted to Android app development.
>
> Thanks for the suggestion. To better reflect the scope of our study, we propose to change the title to “From Assistant to Independent Developer — Are GPTs Ready for Android Software Development?” According to the title change policy of ICLR 2026, we will propose to change after the discussion period.
>
> > *Q1*: **Results in Table 1**: Are the results under “with Compilation Error Feedback” obtained from a single iteration or multiple rounds of refinement? For instance, Qwen3-Coder reports 85.15% compilation success, but this seems inconsistent with the values in Figure 5. In Figure 5, on the line for Qwen3-Coder Compile, there seem to be no node values corresponding to 85.15%. Please clarify.
>
> The “with Compilation Error Feedback” numbers in the main table come from a single iteration round in which the original task description and file-structure outline are always re-included. In Figure 5 we instead run five consecutive repair rounds. To stay within the model’s context limit we omit the task description and file-structure outline after the first round, so the later rounds never see the repeated text. This gradual loss of task context, together with normal LLM sampling randomness, explains the small drop from the single-round result (85.15 % compile / 21.45 % test) to the five-round result (80.20 % compile / 17.46 % test).

---

> ### Author Response · Authors · 2025-11-21
> **Response (2/2)**
>
> > *Q2*: **prompt used for iterative generation based on the previous compilation error output.**
>
> The prompt used for iterative refinement is based on prompts listed in B.6 plus a compilation error message. For experiment with only one feedback attempt, we include the task description once more, while for iterative refining process containing more than one feedback attempts, we include only compilation error messages to meet the context window limit.
>
> Specifically, we use the following prompt template in experiment with only one feedback attempt:
>
> ```
> Your generated code can't be successfully compiled! Compilation errors are
> …<compilation error log>…
> Remember, You are an autonomous programmer and you are modifying a default Android app template with empty activity.
> Your code will be compiled with Gradle7.5.1 and run on Nexus_4_API_31.
> The default Android app template "File Structure" is shown below. You can replace or add some files in the templates to implement the app.
> "File Structure":
> |-- app
> | |-- build.gradle
> | --- src
> | |-- main
> | | |-- AndroidManifest.xml
> ...
>
> Your app should implement every feature in "App Features", and we'll test on each of the features. Note that you should pay attention to the resource-id, content-desc, texts and other attributes we provide with corresponding widgets in "App Features" and exactly match the attributes when implementing the widgets.
> "App Features":
> description: An Android app for quickly sharing your current location.
> feature 1: Share your location via ...
> ...
>
> Please return a json string and only a json string of files to be revised and the revised code in following format:
> {
>     "app/src/main/AndroidManifest.xml":...,
>     ...
> }
> ```
>
> We use the following shortened prompt template in experiment with multiple fix attempts:
>
> ```
> Your generated code can't be successfully compiled! Compilation errors are
> …<compilation error log>…
> ```
> (ends without the task description again)

---

> > ### Comment · Reviewer_TWQk · 2025-11-25
> >
> > Thank you for your clarification. I believe most of my concerns have been addressed.

---

### Official Review · Reviewer_Y9mg · 2025-11-01

**Soundness:** 3
**Presentation:** 3
**Contribution:** 3
**Rating:** 6
**Confidence:** 4

**Summary:**

This paper introduces APPFORGE, a new end-to-end benchmark for evaluating LLMs on real-world software development tasks. APPFORGE requires models to develop full Android applications from scratch, given only natural-language specifications, unlike existing code benchmarks, are focus on a single task. Under human expert verification, each task contains a detailed functional description, natural-language test cases, and implementation constraints for comprehensive information to resolve the problem. Finally, they evaluate 12 SOTA models and two code agents to show that currently, LLMs are still far from handling such an end-to-end development task.

**Strengths:**

1. Novel problem formulation. APPFORGE proposes code evaluation by shifting from “code generation” to “full software engineering.” It evaluates end-to-end development — design, implementation, integration, and runtime — not just isolated snippets.

2. Complete and reproducible evaluation pipeline. The benchmark provides a fully automated, Dockerized workflow: natural-language task → JSON output → Gradle build → emulator test → fuzz evaluation. Both contain static and dynamic evaluation in multiple dimensions.

3. Realistic and validated data construction. Tasks are grounded in real F-Droid apps, with automatic conversion by LLM and human-in-loop verification to guarantee the quality of the tasks. Also, the tasks contain many aspects and kinds of sub-tasks that are highly related to realistic development problems.

**Weaknesses:**

1. The JSON format output is not flexible enough for evaluation. It is understandable that this work is mainly focused on evaluating LLM rather than agents, and such an evaluation method can avoid a lot of environmental problems. But for the SOTA models, agentic tool call and planning are important capabilities, which should be useful and need to be evaluated. Maybe can provide more evaluation for agentic workflows with different models.

2. Currently, generating code based on JSON format may lead to the model needing to output a large amount of code at once when the task is complex. Long outputs may cause a decrease in the quality of the generated code. However, it is uncertain whether this potential problem can be solved by using an agent, COT, or other multiple polling methods. Does the author have any experiments or ideas on that? And it might be good to show the time-consuming and cost for agent evaluation.

**Questions:**

1. It might be good to cite some secure code generation work as a shared related interest for LLM coding.

[1] Seccodeplt: A unified platform for evaluating the security of code genai

[2] SafeGenBench: A Benchmark Framework for Security Vulnerability Detection in LLM-Generated Code

[3] CodeLMSec benchmark: Systematically evaluating and finding security vulnerabilities in black-box code language models

2. It might be good to cite more code agent works as a shared related solution for LLM coding.

[4] OpenHands: An Open Platform for AI Software Developers as Generalist Agents

[5] PatchPilot: A Cost-Efficient Software Engineering Agent with Early Attempts on Formal Verification

[6] Agentless: Demystifying llm-based software engineering agents

3. For future work, it might be exciting to extend to other ecosystems, like iOS, web or desktop. Because different scenarios might involve different programming languages and unique challenges for LLM.

---

> ### Author Response · Authors · 2025-11-21
>
> We sincerely thank you for the insightful and constructive feedback. We address key questions and concerns as below. Please kindly let us know if you have further questions or concerns.
>
> > ​*W1*: **more evaluation for agentic workflows with different models:** Though understandable, for the SOTA models, agentic tool call and planning are important capabilities, which should be useful and need to be evaluated.
>
> We completely agree that agentic capabilities are crucial for SOTA models in software development tasks. We evaluated two code agents (Section 5.1) with command-line use and code execution capabilities. To conduct more evaluation for agentic workflows, we try qwen3-backboned Qwen Code [1], a popular agentic coding agent natively supports web search, and the results are shown below.
>
> | Metric   | Value |
> |------|------|
> | compile | 0.97 |
> | test | 0.18 |
> | success | 0.06 |
>
> Overall, though coding agents are empowered with multi-step planning and tool calls, iterative file creation and modification, and error feedback loops, they achieve marginal improvements compared to the baseline models and fall behind frontier models (11.88%/6.93%/6% vs GPT-5: 18.81%). We will evaluate more agentic systems in revision should the paper be accepted.
>
> *[1] Qwen. (2025, November 15). Qwen Code. https://github.com/QwenLM/qwen-code*
>
>
> > ​*W2*: **performance degeneration potentially caused by JSON format:** Long outputs may cause a decrease in the quality of the generated code. However, it is uncertain whether this potential problem can be solved by using an agent, COT, or other multiple polling methods. Does the author have any experiments or ideas on that? And it might be good to show the time-consuming and cost for agent evaluation.
>
> Thank you for your interesting suggestion! We choose JSON format as most SOTA models (GPT-5, Claude, Gemini) support structured JSON output natively, and we use the same setting for all models to ensure fair comparison. In addition, we evaluated two code agents (Section 5.1) with Qwen3Coder, and the evaluation results do not show improvement with different output formats. We count the token usage and time consumption for Qwen3Coder-backboned mini-swe-agent, which consumes over 34 million tokens in about 9 hours, 30 times more than the tokens consumed using JSON format.
>
> To test whether agentic systems can further improve the performance, we enable the widely-used web search tool and COT for Qwen Code [1], a popular agentic coding agent natively supports web search, and the results is shown below.
>
> | Metric   | Value |
> |------|------|
> | compile | 0.97 |
> | test | 0.18 |
> | success | 0.06 |
>
> Evaluation results demonstrate a success rate of 6%, which demonstrates a marginal improvement.
>
> > ​*Q1/Q2*: **citation of secure code generation and code agent work.**
>
> Thank you for suggesting these relevant citations. We will add them to Section 2 (Related Work) and detailed discussion of related work in the appendix A.
>
> > *​Q3*: **extension to other ecosystems.**
>
> We strongly agree that this is an exciting and valuable direction for future work! We will expand a dedicated Future Work section to include a discussion on extension to multi-platform development. AppForge currently focuses on Android to ensure evaluation reproducibility and automated testing infrastructure. However, the methodology and pipeline we developed are designed to generalize to other ecosystems. Potential platform extensions include iOS with GUI-based apps using Swift/SwiftUI (though requiring macOS infrastructure for Xcode compilation and iOS Simulator testing), web applications using JavaScript/TypeScript/React with Playwright or Selenium-based automated testing, and desktop applications using cross-platform frameworks like Electron or Qt, or native applications with UI automation tools. Each ecosystem presents unique technical challenges that would test different LLM capabilities. iOS requires Apple-specific APIs (UIKit, SwiftUI), stricter app sandboxing and security models, and memory management with ARC. Web applications involve asynchronous operations (Promises, async/await), cross-browser compatibility, client-server architecture, and responsive design. Desktop applications have diverse UI frameworks, OS-specific features (file system access, system tray), and native platform integration.
>
> We believe that AppForge provides a strong foundation and reusable pipeline for developing multi-platform benchmarks. Our automated task construction approach for extracting specifications from real applications and generating test cases can be adapted to other platforms. The human expert validation process (task description review, test case verification) is platform-agnostic. The core evaluation workflow (specification to code generation to compilation to automated testing to fuzzing) can be replicated with platform-specific tools. Our multi-dimensional evaluation applies across platforms.

---

> > ### Comment · Reviewer_Y9mg · 2025-11-25
> >
> > Thanks for the rebuttal, and I think it resolved most of my questions. I keep a positive attitude towards this paper. Thanks!

---

### Official Review · Reviewer_Fhv9 · 2025-11-02

**Soundness:** 3
**Presentation:** 4
**Contribution:** 3
**Rating:** 8
**Confidence:** 4

**Summary:**

The paper presents a benchmark for tackling end-to-end app development. This provides a basis for assessing the abilities of models to be independent developers of complex applications and not just individual aspects of software engineering cycle.

The benchmark is a collection of 101 Android applications sourced from F-droid, the open source Andriod repository. An interesting question in building this is that of how does one obtain the specifications for the task. The paper makes use of application traces from the original apps and then uses LLMs to synthesize test cases from which LLMs generate app description.  The evaluations consist of test cases thus generated. The entire set of tasks were manually verified and validated by software developers.

The evaluations demonstrate that the tasks are challenging for current state-of-the-art models and coding agents.

**Strengths:**

- The benchmark targets end-to-end application development. Relative to existing benchmarks this presents a new investigation, one that is also likely impactful.

- The benchmark creation process is sound. The apps are sourced from real applications and thus represent real needs and the levels of complexity expected in real apps.

- The evaluations test a wide range of models and some coding agents. The substantially low performance numbers indicate the difficulty of this task and provide a useful empirical point on the capabilities of LLMs for end-to-end software engineering. The analysis are generally sound and provide detailed insights into the failure modes and in some cases posits reasons for why. These provide useful insights for future work on this dataset.

**Weaknesses:**

The generation process of the target task descriptions and the test cases are well motivated. I also see that the task descriptions and the test cases have been validated. However, I see some areas of improvement here.
- First,  it is not clear why this would correspond to how users would want to provide descriptions of apps. It would help to motivate how the authors envision such a system being used and how app development specifications are produced.

- Second, the Android Developer Validation section can be improved in terms of details. Currently, it outlines the kinds of checks being done on the task descriptions and the test cases. However, it leaves out some details on what happened when developers found errors, inconsistencies, or completeness problems. Did they modify the descriptions manually? Did multiple developers iterate over each others validation process? Was there any specific validation that checked whether the test cases were clearly aligned with the task descriptions?

- In practice, apps are rarely developed with full functionality in one go, and specifications themselves are often refined in the process of development. Is it reasonable to expect models to develop the entire functionality in one attempt? The process outlined allows for iterative development of functionality in the sense that failures are allowed to be corrected. I wonder if it will be useful to test, if feasible, incrementally add functionalities to the app. At the very least some acknowledgement of how software engineering for apps operates in practice and what differences is being posed in the current setting would be helpful.

**Questions:**

See questions in weaknesses above.

---

> ### Author Response · Authors · 2025-11-21
> **Response (1/2)**
>
> We sincerely thank you for the insightful and constructive feedback and your appreciation of our work as impactful, sound, and providing useful insights. We address key questions and concerns as below.
>
>
> > ​*W1*: it is not clear why this would correspond to how users would want to provide descriptions of apps. It would help to motivate how the authors envision such a system being used and how app development specifications are produced.
>
>
> Thanks for the insightful and constructive suggestion!
>
> We envision that AppForge evaluates AI systems in multiple practical scenarios where users need to provide description to ask models to generate an app from scratch:
> 1. **Rapid Prototyping**: Designers provide high-level descriptions (similar to our task descriptions) to generate initial prototypes, which they use for demonstrating the usefulness or fast iteration. This mirrors how designers communicate with developers, and how developers currently use tools like GitHub Copilot, but at the application level.
> 2. **Automated Code Migration**: Converting legacy apps or porting between platforms where functional specifications exist but implementations need regeneration.
> 3. **Educational & Research Tool**: Training ground for improving LLMs' software engineering capabilities and general intelligence.
> We develop app specifications to mimic these scenarios. Specifically, our task descriptions are derived from *actual app documentation* and *behavior*, representing how developers communicate app requirements in README files, user stories, and design documents. We intentionally balance completeness (providing sufficient detail for implementation) with naturalistic language (avoiding overly prescriptive specifications that would trivialize the task) to mimic real-world users.
> We will add the above illustration to our paper in revision to better motivate our benchmark.
>
>
>
> > ​*W2*: Android Developer Validation section can be improved in terms of details. It leaves out some details on what happened when developers found errors, inconsistencies, or completeness problems. Did they modify the descriptions manually? Did multiple developers iterate over each others validation process? Was there any specific validation that checked whether the test cases were clearly aligned with the task descriptions?
>
> To ensure quality control, five expert Android developers validate every task in two sequential passes and alignment between test cases and task descriptions is explicitly enforced. During the first validation pass, the reviewer performs a line-by-line mapping that requires every test assertion to originate from an explicit statement in the task description; mismatches are flagged and logged. The second, independent reviewer then repeats this mapping exercise and any disagreements were resolved through discussion. A task is accepted only after both reviewers confirm that the final test suite completely and exclusively covers the functionality specified in the description. When problems are found, they either manually clarify the description (8 tasks) by adding missing UI-layout or error-handling constraints, or re-aligned test cases with the description (6 tasks). The Inter-developer agreement rate is 89.1% before discussion. Each task takes 20 minutes on average for manual validation.  The process continues until consensus is reached, yielding benchmarks that mirror authentic Android challenges.  We will update the description in our revision and add an illustrative example in the Appendix B.8.

---

> ### Author Response · Authors · 2025-11-21
> **Response (2/2)**
>
> > *​W3*: **iterative or incremental development scenarios:** In practice, apps are rarely developed with full functionality in one go, and specifications themselves are often refined in the process of development. Is it reasonable to expect models to develop the entire functionality in one attempt? The process outlined allows for iterative development of functionality in the sense that failures are allowed to be corrected. I wonder if it will be useful to test, if feasible, incrementally add functionalities to the app. At the very least some acknowledgement of how software engineering for apps operates in practice and what differences is being posed in the current setting would be helpful.
>
> We conduct pioneering experiments on iterative refinement with failure correction and incremental development with functionality expansion. *In summary, current AI systems achieve limited gains in terms of functional correctness in the above scenarios, compared to the initial one-shot setting.* Details are illustrated below.
>
> We agree with the reviewer that both development in one go and iterative/incremental development are important and common software engineering practices. In revision, we will acknowledge these two software engineering practices, provide details on the pioneering experiments below, and discuss future avenues for extending our benchmark in the iterative/incremental development settings.
>
> **Experiments on iterative refinement with failure correction**:
> In Section 4.2, we described an evaluation of the model’s performance in terms of iterative refinement with compilation feedback. The compilation error feedback substantially improves compilation success across all models, with notable improvements for Claude-4-Sonnet (40.59% to 77.23%) and Qwen3-Coder (27.72% to 85.15%). However, this improvement does not translate proportionally to functional correctness, as test pass rates show modest gains. The Figure 5 in our paper demonstrates the performance evolution with compilation feedback.
>
> **Experiments on incremental development with functionality expansion**:
> We make an experiment by splitting our specifications into single feature tasks for qwen3-backboned Claude Code, but the results showed a significant performance drop:
>
> | Metric   | Value |
> |----------|-------|
> | compile  | 0.51  |
> | test     | 0.07  |
> | all_pass | 0.02  |
>
> We hypothesize that incremental development in Android is challenging. Android codebases typically have higher coupling and dependency, meaning that adding a new feature often requires modifying a broader scope of the existing codebase, not just inserting new code. We will provide more details and test more systems in revision.

---

### Meta-Review · Area_Chair_tuuE · 2026-01-08

**Summary:**

The authors proposed AppForge, a new benchmark to evaluate LLMs for implementing complete software (Android apps) totally from scratch. The benchmark bridges the gap from current benchmarks by challenging LLMs to understand and coordinate throughout the entire development cycle to generate context-aware, robust, and maintainable code. The authors designed a multi-agent system to generate test cases for automatic evaluation framework. They evaluated 12 LLMs and showed that most models performed poorly in developing complex software engineering tasks.

**Reviewer Concerns:**

- There are some concerns about the generation of the target tasks and the test cases e.g. how users want to provide descriptions of apps, the validation by Android developer
    - To address these, the authors responded to explain their motivation to create practical scenarios where users need to provide descriptions of apps. They also explained clearly the validation process to ensure quality control
- The JSON format output is not flexible for evaluation and may cause issues during generation when the model needs to generate a large amount of code.
    - To address these concerns, the author added new results with agentic methods
- The model behaviours e.g. GPT4.1 evades development by deleting faulty code, were not clearly explained and the root cause was not explored enough
    - The authors added additional case analysis and inspection of prompts. While I think this response is a bit post-hoc and not a systematic study, it does address the reviewer specific questions for GPT model behaviour.
- There is some confusion in the title of the paper and what the real contributions are, including what abilities are really being tested on LLMs.
    - The authors tried to justify their motivation and define their scope in the responses. While I still acknowledged the novelty and significant contribution, I recommend the authors clearly justify their claims accordingly to the real scope in the paper.
- Data leakage concerns because the source code of popular apps might be released publicly.
    - The authors conducted analysis to emphasise the generated code is very different from the real code, implying that data is not leaked to the models. They also explained their generation pipeline such that the generated tasks are different from the original source implementation.

**Reviewer Scores:**

- Reviewer Fhv9 would keep the positive score of 8
- Reviewer Y9mg would keep the positive score of 6
- Reviewer TWQk may increase the score from 6 to 7 given their concerns are addressed
- Reviewer WwYL might increase their score from 4 to 5 as their major concerns are sufficiently addressed

---

### Decision · Program_Chairs · 2026-01-26

Accept (Poster)